# THE EFFECT OF TEMPORAL RESOLUTION IN OFFLINE TEMPORAL DIFFERENCE ESTIMATION

## ABSTRACT

Temporal Difference (TD) algorithms are the most widely employed methods in Reinforcement Learning. Notably, previous theoretical analysis on these algorithms consider the sampling time as fixed a priori, while it has been shown that the temporal resolution can impact data efficiency (Burns et al., 2023). In this work, we analyze the performance of mean-path semi-gradient TD(0) for offline value estimation, emphasizing the dependence on the temporal resolution, a factor that indeed proves to be of crucial importance. For continuous-time stochastic linear quadratic dynamical systems with a fixed data-budget, the Mean Squared Error in value estimation shows an optimal non-trivial value for the time discretization, and this choice affects the reliability of the algorithm. We also show that this behavior differs from that of the Monte Carlo algorithm (Zhang et al., 2023). We verify the theoretical characterization in numerical experiments in linear quadratic system instances and further demonstrate, in a stochastic control setting, that the step-size trade-off persists in policy iteration.

## 1 INTRODUCTION

Temporal Difference (TD) is a fundamental idea in Reinforcement Learning (RL) based on bootstrapping value estimates from sampled rewards and current predictions, and it has nowadays become the core method for model-free reinforcement learning algorithms. In RL, samples typically come from a sampling procedure which follows discrete time intervals, where the temporal resolution is fixed a-priori for each application. Previous studies have shown that temporal resolution is an important factor in data efficiency (Burns et al., 2023; Zhang et al., 2023) but is often overlooked in RL research. While the convergence and statistical properties of TD have been studied extensively in the literature (Sutton, 1988; Jaakkola et al., 1993; Tsitsiklis & Van Roy, 1997; Bhandari et al., 2018; Lakshminarayanan & Szepesvari, 2018; Asadi et al., 2024), little is known about the effect of temporal discretization on the TD algorithm from both theoretical and applied perspectives.

In this paper, we study the impact of temporal resolution in value estimation using TD. In particular, we look into a specific class of systems, a continuous-time linear stochastic dynamical system with quadratic instantaneous reward (see e.g. Zhang et al. (2023)):

$$\begin{cases} \mathrm{d}x(t) = ax(t)\mathrm{d}t + \sigma \mathrm{d}w(t) \\ V(x(\tau)) = -\mathbb{E}[\int_\tau^\infty \gamma^{t-\tau} q x^2(t)\mathrm{d}t] \end{cases} \tag{1}$$

where $w(t)$ is a Wiener process. The drift coefficient $a$ is unknown, while the diffusion coefficient $\sigma$, the reward weight $q$ and the discount factor $\gamma \in (0,1)$ are assumed to be known. The value function $V(\cdot)$ is defined as the expected cumulative discounted reward. Estimating the infinite-horizon value $V(x(\tau))$ corresponds to policy evaluation for a fixed linear policy in the continuous-time Linear Quadratic Regulator (LQR) (Lindquist, 1990; Zhang et al., 2023). Note that the optimal policy for this problem is indeed linear in the state. We analyze the Mean-Squared Error (MSE) of the value estimate from a widely used TD algorithm, semi-gradient TD(0) (Sutton & Barto, 2018), in the offline setting, in order to understand how finite-sample properties change with respect to the temporal resolution. By leveraging the fact that for this specific type of system, we can compute the $n$-th moment of the state in closed form, for any $n$, we provide a characterization of the MSE and identify a trade-off modulated by temporal resolution. Fig. 1 illustrates the trade-off through a numerical experiment, where we plot the learning curve of an offline mean-path semi-gradient

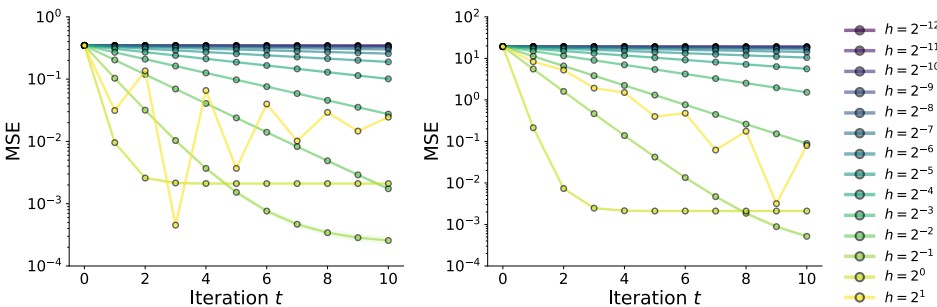

Figure 1: Learning curves of TD(0) show different behavior with respect to temporal resolution $h$.

TD(0) algorithm (Bhandari et al., 2018), under two different initializations (see Appendix A.1). The result shows that the best MSE is achieved at an intermediate temporal resolution $h$, highlighting the existence of a non-trivial optimal discretization.

The contributions of our work are as follows. First, we develop a framework for analyzing the impact of temporal resolution on offline TD value estimation. Second, we derive an approximate expression of MSE for Offline Mean-Path Semi-gradient TD(0), which shows a trade-off with respect to the length of the sampling intervals. We then derive the optimal temporal resolution $h^*$ that shows how it scales with the data budget $B$. In addition, we establish that the MSE scaling retains the same order with respect to the step size under function approximation errors. Furthermore, we identify sufficient conditions for the convergence of offline mean-path TD(0) and provide finite-sample bounds for both online mean-path and stochastic TD(0). These results extend existing analysis of semi-gradient TD(0) to the continuous, unbounded state space of linear quadratic systems. Lastly, we contrast the trade-off with that of Monte Carlo methods and offer suggestions for choosing temporal resolution in practice. We conduct numerical experiments to validate the theoretical findings. We also confirm that the trade-off generalizes to multidimensional systems and the stochastic TD algorithm. We further demonstrate in a stochastic control setting that the step-size trade-off extends to policy learning. To our best knowledge, this work represents a first step toward understanding the impact of the temporal resolution in TD methods.

## 2   RELATED WORKS

**Temporal discretization**   It is well known that the choice of temporal discretization can affect the performance of various RL algorithms. This literature fall into two main categories. The first one studies temporal abstraction, built on top of a base discretization. Sutton et al. (1999) formalized this in the options framework. Numerous variants have shown improved performance, particularly in video games (Sharma et al., 2017; Lakshminarayanan et al., 2017; Machado et al., 2018; Metelli et al., 2020; Dabney et al., 2021). The other line of work is concerned with the base-level discretization rather than building abstractions (Huang et al., 2019; Huang & Zhu, 2020; Park et al., 2021; Lutter et al., 2022; Farrahi & Mahmood, 2023).

The work with the most relevant problem setting to ours is a recent study by Zhang et al. (2023) which analyzed the impact of temporal discretization on the value estimation performance of Monte Carlo methods. Similar to our setting, their work focused on linear quadratic systems and provided analytical results for both finite horizon and infinite horizon settings. However, Monte Carlo methods operate in a fundamentally different way from temporal difference. It remains an open question whether the trade-off observed in their setting extends to TD learning for continuous-time systems.

**Continuous-time RL**   Our work focuses on continuous-time dynamical systems. Although RL typically assumes a discrete-time framework, several works have applied RL to continuous-time systems (Baird, 1994; Bradtke & Duff, 1994; Doya, 2000; Wang et al., 2020; Basei et al., 2022; Jia & Zhou, 2022b). Jia & Zhou (2022a) provides a unified continuous-time formulation of various TD methods, and proved that the time-discretized version of these algorithms converge to the continuous-time counterpart in the limit of the discretization. However, the behavior and estima-

tion error of the discretized TD algorithms with a non-zero discretization $h$, over a continuous state space, have yet to be characterized.

**Theoretical analysis of TD**   Theoretical properties of TD methods have been extensively studied in the literature, as mentioned in Appendix 1. However, we do not revisit them here, since our focus is on understanding how TD value estimation is affected by temporal resolution. Readers interested in a recent overview of TD theory are referred to the related work sections in Tu & Recht (2018) for Least-squares based methods and in Patil et al. (2024) for stochastic-gradient based methods.

In this work, we focus on a specific algorithm of TD known as the mean-path semi-gradient TD(0), in the offline setting. Semi-gradient TD(0), a standard member of the TD family, updates parameters by following the semi-gradient of the squared TD-error with respect to the parameter (Sutton & Barto, 2018). The mean-path version, introduced by Bhandari et al. (2018), instead follows the mean negative semi-gradient under the stationary distribution. Their finite-sample analysis for mean-path TD did not account for time discretization, nor provided closed-form expressions for estimation quality — both of which are crucial for trade-off analysis. Furthermore, they assume that data are sampled directly from the stationary distribution. This ignores the transient dynamics inherent in practical settings, where data collection includes the mixing phase. However, this algorithm serves as a good starting point for our analysis. Relatedly, Xiao et al. (2021) analyzed the fixed-point of offline semi-gradient TD(0), under finite state space and overparameterized function approximation, which differs from our setting. And they did not consider time discretization.

## 3   PROBLEM SETTING

In this section, we describe the setting where the analysis will be performed, namely, the system, the data, the algorithm, and the objective.

### 3.1   CONTINUOUS-TIME STOCHASTIC LINEAR QUADRATIC SYSTEM

As discussed in Appendix 1, the dynamics and the return of the system are given by Eq.1. Without loss of generality, we set the weight of the reward $q = 1$ and assume that the process starts at $x(0) = 0$ (Abbasi-Yadkori et al., 2011; Dean et al., 2020; Zhang et al., 2023). To ensure the value $V \in \mathbb{R}$ is finite, we assume $a < 0$. Using Lemma A.1 from (Zhang et al., 2023), we can derive the closed-form expression for the value $V$ at $x(0)$:

$$V := V(x(0)) = \int_0^\infty \frac{\gamma^t \sigma^2}{2a} \left(1 - e^{2at}\right) \, \mathrm{d}t = \frac{-\sigma^2}{(\ln \gamma)(\ln \gamma + 2a)} \tag{2}$$

We consider a linear function approximation of the value function parameterized by $\theta$: $V_\theta(x) = \phi(x)\theta$, where the value is linear in the feature $\phi(x)$. We follow Tu & Recht (2018) and choose the feature as $\phi(x) := x^2 - \frac{\sigma^2}{\ln \gamma}$. Since the value function of a linear quadratic system is quadratic in the state $x$, it lies exactly in the span of the features. In particular, at the initial state, we have $V_\theta(0) = \phi(0)\theta = -\frac{\sigma^2}{\ln \gamma}\theta$. Equating with Equation 2 gives the true parameter: $\theta^* = \frac{1}{\ln \gamma + 2a}$.

### 3.2   OFFLINE DATASET SAMPLED AT TIME INTERVAL $h$

We work with offline data sampled from the continuous-time dynamics described by Equation 1 at discrete time. The dynamics are sampled $N$ times per trajectory, under a finite data budget $B$. The data collection procedure is identical to the one in Zhang et al. (2023), where data are sampled through a uniform discretization of the interval $[0, T]$, with $T < \infty$ being the *estimation horizon*, with time increment $h$. This results in the collection of $N = T/h$ points (which for simplicity is assumed to be an integer) over a single trajectory, at times $t_k := kh$, for $k = 0, \ldots, N - 1$. Given the data budget $B$, it is therefore possible to sample from $M = B/N$ different trajectories. At each time instant $t_k$ of each trajectory $i$, the state $x_i(t_k)$ is observed and the approximate reward incurred in the interval $[t_k, t_k + h]$ is computed as $r_i(t_k) = -hx_i^2(t_k)$. The offline dataset is gathered as $\mathcal{D} = \{(x_i(t_k), r_i(t_k), x_i(t_{k+1})) \mid i = 1, 2, \ldots, M \text{ and } k = 0, 1, \ldots, N - 2\}$.

We focus on the offline setting to strictly decouple the sample size from the number of algorithmic updates – quantities that are intrinsically coupled in online learning. By fixing the dataset, we isolate

### 3.3 MEAN-PATH SEMI-GRADIENT TD(0) ON OFFLINE DATA

The semi-gradient TD(0) algorithm starts with an initial parameter estimate $\theta_0$, which gets updated iteratively toward the true parameter $\theta^*$. At iteration $t$, it updates the current estimate $\theta_t$ according to the sampled triplet containing current state, reward and next state $(x, r, x')$, by $\theta_{t+1} = \theta_t + \alpha g_t(\theta_t)$ where $\alpha$ is the learning rate, and $g_t(\theta_t)$ is the negative semi-gradient at iteration $t$: $g_t(\theta_t) = \left(r + \left(\gamma^h \phi(x') - \phi(x)\right) \theta_t\right) \phi(x)$, where $\gamma^h$ is the effective discount factor in the discretized system. In this work, we consider instead an *offline* version of the mean-path TD introduced by Bhandari et al. (2018), whose update rule involves the mean negative semi-gradient over some distribution rather than the stochastic gradient. In the offline setting, the mean negative semi-gradient is computed over the empirical distribution induced by the whole dataset $\mathcal{D}$, collected according to the procedure described in Appendix 3.2. The update rule is hence

$$\theta_{t+1} = \theta_t + \alpha \bar{g}(\theta_t), \tag{3}$$

where the mean of the negative semi-gradient is

$$\bar{g}(\theta_t) = \overline{\phi r} + \overline{\phi(\gamma^h \phi' - \phi)}\theta_t$$

$$= \frac{1}{M(N-1)} \sum_{i=1}^{M} \sum_{k=0}^{N-2} \phi\left(x_i(t_k)\right) \left(r_i(t_k) + \left(\gamma^h \phi\left(x_i(t_{k+1})\right) - \phi\left(x_i(t_k)\right)\right) \theta_t\right), \tag{4}$$

where $\overline{\phi r}$ and $\overline{\phi(\gamma^h \phi' - \phi)}\theta_t$ are shorthands denoting taking the mean over the triplet $(\phi, r, \phi')$ in the dataset. As a deterministic counterpart to stochastic TD, mean-path analysis avoids the worse-case upper bounds common in the literature (Bhandari et al., 2018). Instead, we show in Appendix 4 that it yields closed-form expressions that capture the exact error landscape, revealing trade-offs that is otherwise obscured by conservative concentration bounds. Furthermore, this formulation transforms the time-dependent stochastic updates into a time-invariant setting, which is crucial in the analysis as it isolates the statistical moments of the data from the algorithmic updates.

### 3.4 OBJECTIVE: MEAN-SQUARED ERROR OF VALUE ESTIMATION

We characterize the Mean-Squared Error of the value estimate from the offline mean-path semi-gradient TD(0) algorithm described above. It is a function of the parameter estimate $\theta_t$ after $t$ updates: $\text{MSE}_t = \mathbb{E}\left[(V_{\theta_t} - V)^2\right]$ where $V_{\theta_t}$ and $V$ are the infinite-horizon value estimate after $t$-step updates and the true value, respectively. $V_{\theta_t}$ is determined by the parameters $h, B, T, \sigma, \alpha, \theta_0, t$.

## 4 THEORETICAL RESULTS ON SEMI-GRADIENT TD(0)

The main goal of this section is to gather insights on the behaviour of the MSE with respect to the temporal resolution parameter $h$, through the analysis of the evolution of the parameter $\theta_t$. Recall that the ground truth value is $V = -\frac{\sigma^2}{(\ln \gamma)(\ln \gamma + 2a)}$. With $t$ step update with the semi-gradient, we have the value estimate $V_{\theta_t} = -\frac{1}{\ln \gamma}\sigma^2 \theta_t$. The corresponding MSE can be expressed as follows:

$$\text{MSE}_t = \mathbb{E}\left[(V_{\theta_t} - V)^2\right] = \frac{\sigma^4}{(\ln \gamma)^2} \left(\mathbb{E}[\theta_t^2] - \frac{2\mathbb{E}[\theta_t]}{\ln \gamma + 2a} + \left(\frac{1}{\ln \gamma + 2a}\right)^2\right), \tag{5}$$

where the expectation is taken w.r.t. the distribution of the data generated by the process $x(\cdot)$.

### 4.1 MSE FOR OFFLINE MEAN-PATH SEMI-GRADIENT TD(0)

The following theorem provides the characterization of the MSE for Offline Mean-Path Semi-gradient TD(0) after $t$ updates, provided the discretization step-size is small: $h \in (0, 1)$.

**Theorem 4.1** (Mean Squared Error). *After $t$ updates, the mean squared error is*

$$\text{MSE}_t = \frac{\sigma^4}{(\ln \gamma)^2} \Bigg\{ \left[ t^2\alpha^2\mathcal{I}_3 + 2t\alpha\theta_0\left(\mathcal{I}_1 + (2t-1)\alpha\mathcal{I}_5\right) + \theta_0^2\left(1 + 2t\alpha\mathcal{I}_2 + t(3t-2)\alpha^2\mathcal{I}_4\right)\right]$$

$$- \frac{2}{\ln\gamma + 2a}\left[\theta_0 + t\alpha(\mathcal{I}_1 + \mathcal{I}_2\theta_0) + \frac{t(t-1)}{2}\alpha^2\left(\mathcal{I}_5 + \mathcal{I}_4\theta_0\right)\right] + \left(\frac{1}{\ln\gamma + 2a}\right)^2 \Bigg\} + \mathcal{O}(h^3) \quad (6)$$

*where $\mathcal{I}_1, \cdots, \mathcal{I}_5$ are auxiliary terms dependent on $h$ but not $t, \alpha$, introduced in Appendix A.2. Importantly, the MSE can be expressed as:*

$$\text{MSE}_t = C_0 + C_1 h + C_2 h^2 + \mathcal{O}(h^3) \quad (7)$$

*where $C_0 \geq 0, C_1 \leq 0, C_2 \geq 0$ are constants with respect to $h$, given by:*

$$C_0 = \frac{\sigma^4}{(\ln\gamma)^2}\left(\theta_0 - \frac{1}{\ln\gamma + 2a}\right)^2,$$

$$C_1 = \frac{t\alpha\sigma^4}{(\ln\gamma)^2}\left(\theta_0 - \frac{1}{\ln\gamma + 2a}\right)^2\left[-2\left(2a + \ln\gamma\right)C_{11} + \frac{\alpha(2t-1)\left(2a + \ln\gamma\right)^2 C_{31}}{B}\right],$$

$$C_2 = \frac{t\alpha\sigma^4}{(\ln\gamma)^2}\left(\theta_0 - \frac{1}{\ln\gamma + 2a}\right)^2\Big[2C_{23} - 2(2a + \ln\gamma)C_{12} +$$

$$(C_{11}^2 + \frac{C_{320}}{B})(2a + \ln\gamma)^2(2t-1)\alpha\Big].$$

*The constants $C_{11} < 0, C_{12} > 0, C_{23} > 0, C_{31} < 0, C_{320} > 0$ depend only on $a, T, \ln\gamma, \sigma^4$, and their precise forms are given in Appendix A.2.*

The theorem presents the expression for the $t$-step MSE in Equation 6. In order to clearly exhibit the order of $h$ in the MSE, we derive another approximate form of $t$-step MSE in Equation 7, offering more interpretable insights. For small $h$, the MSE approximately follows a quadratic relation in $h$, and the minimum is attained when $h$ is strictly positive, i.e., $h^* > 0$. It confirms the existence of a trade-off in the temporal resolution parameter for the offline mean-path semi-gradient TD(0).

## 4.2 OPTIMAL TEMPORAL RESOLUTION $h^*$ IN OF OFFLINE MEAN-PATH TD(0)

The optimal discretization step-size $h^*$ represents the time interval at which we would ideally sample our dynamical system in order to have the best estimation of the value in term of the MSE. A precise form for this optimal parameter can be found by exploiting the approximate expression of the MSE in Equation 7, as shown in the next corollary.

**Corollary 4.2** (Optimal Discretization). *The optimal $h^*$ based on the approximation Equation 7 after $t$ updates is*

$$h^* \approx -\frac{C_1}{2C_2} = -\frac{-2\left(2a + \ln\gamma\right)C_{11} + \frac{\alpha(2t-1)(2a+\ln\gamma)^2 C_{31}}{B}}{2\left[2C_{23} - 2(2a + \ln\gamma)C_{12} + (C_{11}^2 + \frac{C_{320}}{B})(2a + \ln\gamma)^2(2t-1)\alpha\right]}, \quad (8)$$

*and the minimum MSE is*

$$\text{MSE}_t^* \approx \frac{\sigma^4\left(\theta_0 - \frac{1}{\ln\gamma + 2a}\right)^2}{(\ln\gamma)^2}$$

$$\left[1 - \frac{4t\alpha\alpha\left(-2\left(2a + \ln\gamma\right)C_{11} + \frac{\alpha(2t-1)(2a+\ln\gamma)^2 C_{31}}{B}\right)^2}{2C_{23} - 2(2a + \ln\gamma)C_{12} + (C_{11}^2 + \frac{C_{320}}{B})(2a + \ln\gamma)^2(2t-1)\alpha}\right]. \quad (9)$$

The expression in Equation 8 is clearly dependent on the specific dynamical system or environment at hand. Therefore setting the time discretization to the optimal value would be impossible without full knowledge of the dynamics. Although it is possible to empirically find the optimal temporal

resolution by sweeping over different discretization intervals, it would be impractical to sample the dataset at different frequencies just to maintain the one that has proved the most effective in terms of the MSE for the value estimation. On the other hand, if the $1/B$ terms are relatively small, the resulting optimal $h$ would be insensitive to the change in $B$. We will show empirically in Appendix 5 that it is indeed the case.

For large enough data budgets $B$, we can show that the optimal time discretization $h^*$ is independent from the data budget, and further simplify the expressions, shown in the next corollary.

**Corollary 4.3** (Asymptotic Optimal Discretization). *(i) If the budget $B$ is large while the horizon $T$ is fixed and finite, one can obtain*

$$\text{MSE}_t = \left\{ 1 + t\alpha \left[ -2\left(2a + \ln\gamma\right)\left(C_{11}h + C_{12}h^2\right) + 2C_{23}h^2 + C_{11}^2 h^2 (2a + \ln\gamma)^2 (2t - 1)\alpha \right] \right\}$$

$$* \frac{\sigma^4}{(\ln\gamma)^2} \left( \theta_0 - \frac{1}{\ln\gamma + 2a} \right)^2 + \mathcal{O}(\frac{1}{B}) + \mathcal{O}(h^3).$$

$$h^* \approx -\frac{-2\left(2a + \ln\gamma\right)C_{11}}{2\left[2C_{23} - 2(2a + \ln\gamma)C_{12} + C_{11}^2 (2a + \ln\gamma)^2 (2t - 1)\alpha\right]}.$$

*(ii) If the horizon $T$ is large (and thus $B$ is also large, since $B = \frac{TM}{h}$), we have*

$$\text{MSE}_t = \frac{\sigma^4}{(\ln\gamma)^2} \left( \theta_0 - \frac{1}{\ln\gamma + 2a} \right)^2 \left\{ 1 + t\alpha \left[ \frac{\sigma^4 (2a + \ln\gamma)(2a + 3\ln\gamma)}{2a^2 \ln\gamma} h \right. \right.$$

$$\left. \left. + (2a + \ln\gamma)^2 \left( \frac{3\sigma^4}{4a^2} + \frac{\sigma^8 (2a + 3\ln\gamma)^2 (2t - 1)\alpha}{16a^4 (\ln\gamma)^2} \right) h^2 \right] \right\} + \mathcal{O}(\frac{1}{T}) + \mathcal{O}(h^3).$$

$$h^* \approx -\frac{4a^2 \ln\gamma(2a + 3\ln\gamma)}{(2a + \ln\gamma)\left(12a^2(\ln\gamma)^2 + \sigma^4 (2a + 3\ln\gamma)^2 (2t - 1)\alpha\right)}$$

*Remark* 4.4. The two cases in Corollary 4.3 are consistent: letting $T$ be large in (i) recovers the expression in (ii).

**How to choose temporal resolution for TD** The fact that $h^*$ is insensitive to the data budget $B$ has important practical implications. An optimal $h^*$ can be efficiently determined by performing a grid search on $h$ using a baseline data budget $B_0$. Concretely, we can consider an initial "burn-in" phase: collect a dataset of size $B_0$, estimate the value $V$ via Monte Carlo as in Zhang et al. (2023), and perform a grid search over $h$ based on the empirical MSE, by sub-sampling this dataset. Then increasing $B$ can verify if $h^*$ remains stable. If so, the same $h$ can be reused for larger data budgets, thereby reducing hyperparameter search costs while maintaining accurate value estimation.

### 4.3 MSE OF OFFLINE MEAN-PATH TD(0) UNDER FUNCTION APPROXIMATION ERRORS

Our analysis above assumes that the true value function lies in the linear span of the features. However, the standard setting in RL (Sutton & Barto, 2018) is underparameterized, where the features do not perfectly represent the value function, leading to function approximation errors. We demonstrate that the fundamental characteristics of our analysis remain valid in this regime. Specifically, we show that the scaling of MSE with respect to step-size retains the same order in the underparameterized setting. The detailed derivation and analysis are provided in Appendix A.6.

### 4.4 CONVERGENCE ANALYSIS OF TD(0)

Existing TD(0) analysis (Tsitsiklis & Van Roy, 1997; Bhandari et al., 2018) is limited to finite state spaces and does not apply to the continuous domain of stochastic linear quadratic systems. While the finite-sample behavior of LSTD has been analyzed (Tu & Recht, 2018), we are unaware of any convergence results for TD(0) in this setting. We bridge this gap by establishing the convergence properties of TD(0) in stochastic LQ systems across three algorithmic settings: 1, offline mean-path: we identify sufficient conditions for the convergence to the corresponding LSTD estimate (details in Appendices A.4.1 and A.5); 2, standard mean-path: we provide a finite-sample analysis, generalizing the framework of Bhandari et al. (2018) to stochastic LQ systems (see Appendices A.4.2 and A.5); 3, online stochastic TD: we provide a finite-sample analysis for the online setting, similarly extending the analysis of Bhandari et al. (2018) to the unbounded domain (see Appendix A.7).

### 4.5 COMPARISON WITH MONTE CARLO

Recent work by Zhang et al. (2023) established that Monte Carlo (MC) estimation exhibits a trade-off in MSE w.r.t. $h$, under the same problem setting as ours. They derived the exact MSE expression (Theorem 3.6 in Zhang et al. (2023)) and showed that $\text{MSE}_{\text{MC}} = \mathcal{O}(\frac{1}{hB} + h)$. They further demonstrated that the optimal $h$ scales polynomially with $B$, namely: $h^*_{\text{MC}} \approx B^{-1/2}$. In contrast, our analysis indicates that for TD learning, the optimal step-size $h^*$ behaves differently – it remains largely constant w.r.t. B.

To build intuition, consider how variance reacts to the changes in the data budget $B$. TD implicitly performs a maximum-likelihood fit of the value-function parameters within its chosen model (Sutton & Barto, 2018). Once sufficient data are available to obtain a stable parameter estimate, additional samples yield little further variance reduction. This explains why the trade-off and hence $h^*$ is largely insensitive to $B$. In contrast, the Monte-Carlo estimator in Zhang et al. (2023) directly averages returns. Increasing $B$ continues to reduce trajectory variance, hence affecting the trade-off.

In the next section, we present numerical experiments that illustrate and confirm these theoretical differences between TD and MC estimation.

## 5 NUMERICAL EXPERIMENTS

To empirically validate our theoretical analysis in the previous section, we conduct simulations on continuous-time stochastic linear quadratic systems. While our theoretical framework characterizes the trade-off in Langevin dynamics, we investigate whether these insights hold for TD in practice, especially for multi-step updates. By systematically varying temporal resolution, data budget, and system parameters, we quantify how the discretization choices impact the MSE of the value estimation of TD. We also perform a comparison between TD and Monte Carlo methods.

### 5.1 OFFLINE MEAN-PATH TD ON LINEAR QUADRATIC SYSTEMS

In our experiments, we perform 50 independent runs to approximate the expectation in the MSE computation. In each run, we generate a new dataset by simulating the Langevin process of Appendix 3.1 with a unique random seed, following the procedure outlined in Appendix 3.2. We then apply the offline mean-path semi-gradient TD(0) algorithm, as described in Appendix 3.3, to obtain an estimate and compute the squared error relative to the true value. The lines in the plots represent the mean squared error averaged over the 50 runs, while the shaded regions indicate the standard error. We fix the parameter $\sigma = 1$ throughout the experiments. The values of $h$ is chosen from this grid: $h \in \left(\{2^{-15}, 2^{-14}, \cdots, 2^{-2}\}\right) T$.

**Trajectory and convergence of the iterates:** In order to understand the evolution over updates of the parameter $\theta_t$, when following the gradient dynamics in equation 3, we can start by looking at the fixed points of the latter. If $\bar{\theta}$ is a fixed point of the gradient dynamics, then from equation 3 we have that $\bar{\theta}$ must satisfy $\bar{g}(\bar{\theta}) = 0$. From equation 4 we then derive:

$$\bar{\theta} = -\left(\overline{\phi(\gamma^h \phi' - \phi)}\right)^{-1} \overline{\phi r}$$

$$= -\left(\sum_{i=1}^{M} \sum_{k=0}^{N-2} \phi(x_i(t_k)) \left[\gamma^h \phi(x_i(t_{k+1})) - \phi(x_i(t_k))\right]\right)^{-1} \sum_{i=1}^{M} \sum_{k=0}^{N-2} \phi(x_i(t_k)) r_i(t_k), \quad (10)$$

which represents the unique fixed point, and it coincides with the LSTD estimate $\theta_{\text{LSTD}}$.
Convergence to the LSTD estimate is empirically shown in Figure 2, where the evolution of the parameter $\theta_t$ converges to the unique fixed point, and indeed the average semi-gradient converges to 0. From Figure 2 one can note that $\theta_t$ converges to the LSTD estimate even if it starts closer to the true parameter $\theta^*$, as is the case in plot (b), while convergence to the optimal parameter is achieved only if the latter coincides with $\theta_{\text{LSTD}}$, as shown in plot (d).

**Asymptotic MSE vs $h$:** In Figure 3, we illustrate how the asymptotic MSE varies with $h$, under the parameters $a = -8, T = 8, \gamma = 0.9$. For each $h$, the learning rate is optimized from $\{0.1, 1.0, 10.0\}$ and TD is run until convergence. The plot shows the MSE for three different initializations of $\theta_0$. In all cases, the iterates converge to the LSTD estimate, consistent with the earlier discussion on convergence.

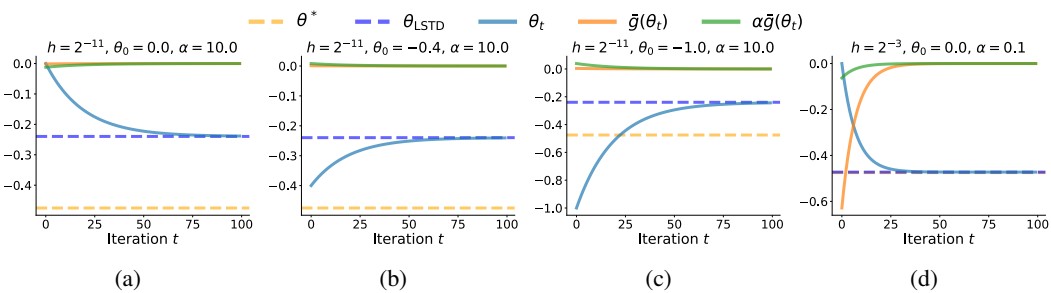

Figure 2: Trajectory of the parameter $\theta_t$ as it converges to the fixed point $\theta_{\text{LSTD}}$

**Dependence of MSE and $h^*$ on the data budget $B$:** We plot the asymptotic MSE of TD as a function of $B$ while keeping other parameters fixed to $a = -8, T = 8, \gamma = 0.9, \theta_0 = 0$. As shown in Figure 5 (left), increasing $B$ generally reduces the MSE, since more data yields more accurate estimates. However, varying $B$ has negligible effect on the optimal step size $h^*$. It aligns with the trend in Figure 13 for one-step TD (Appendix), where $h^*$ remain stable across different $B$.

**MSE for multi-dimensional systems:** To investigate whether the trade-off generalizes from scalar to vector systems, we run experiments on a three-dimensional system where the dynamics matrix $A$ is randomly sampled following the scheme in Zhang et al. (2023). Figure 4 demonstrates that the trade off and the behavior of $h^*$ w.r.t $B$ persists in the vector setting.

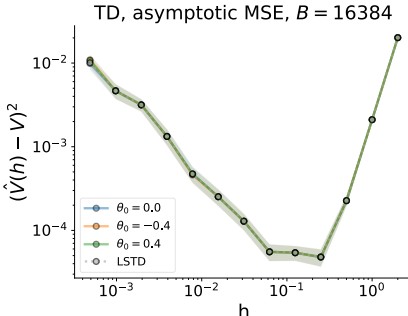

Figure 3: Asymptotic MSE

**MSE under varying dynamics parameter $a$:** Figure 5 (right) illustrates the asymptotic MSE when we vary system dynamics parameter $a$ over $\{-1, -2, -4, -6, -8, -16\}$. The other parameters are fixed to $T = 8, B = 4096, \gamma = 0.9, \theta_0 = 0$. As $|a|$ increases, the MSE across all step sizes $h$ decreases as the system decays faster.

**MSE at various number of updates $t$:** Figure 6 illustrates how the MSE evolves w.r.t $h$ over update steps, under two different algorithm parameter settings while keeping the system parameters fixed at $a = -8, T = 8, B = 16384, \gamma = 0.9$. In both plots, the algorithm is run for 100 update steps for each fixed $h$, with learning rate $\alpha = 0.1$. The left plot starts from $\theta_0 = 0$, while the right plot starts from $\theta_0 = 0.4$. In both cases, the MSE decreases with the number of updates and converges quickly. However, the trade-off in MSE w.r.t $h$ persists as the updates progress. Notably, the optimal step size $h^*$ appears to remain stable once the number of updates $t$ is sufficiently large.

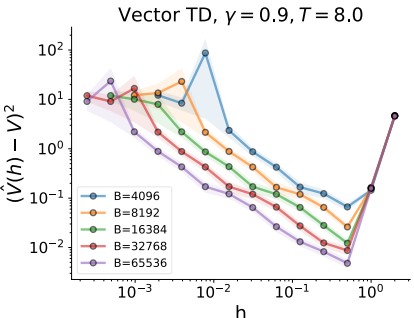

Figure 4: MSE for 3-dimensional systems

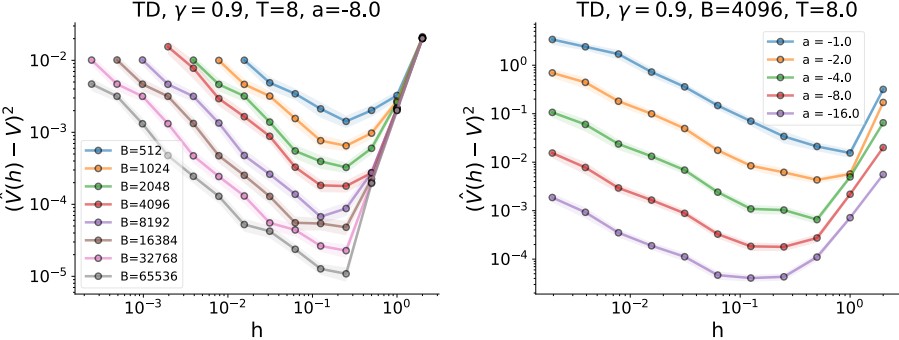

Figure 5: MSE under varying $B$ and $a$, respectively

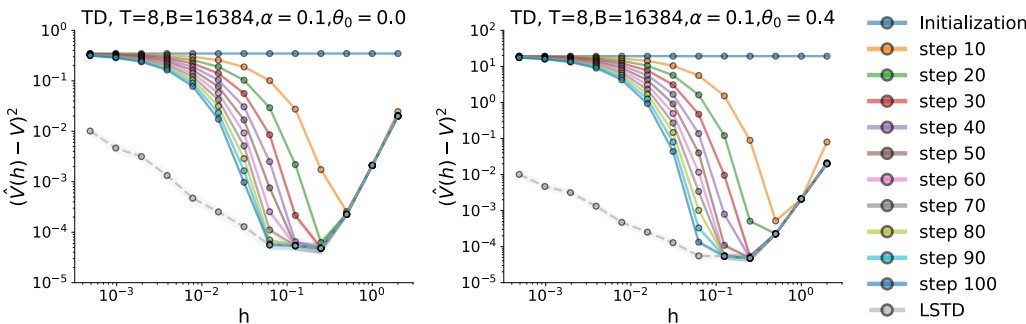

Figure 6: Empirical MSE as a function of $h$ as number of steps increase

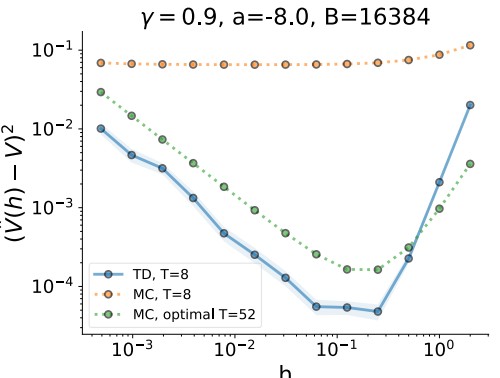

Figure 7: MSE of TD compared with MC

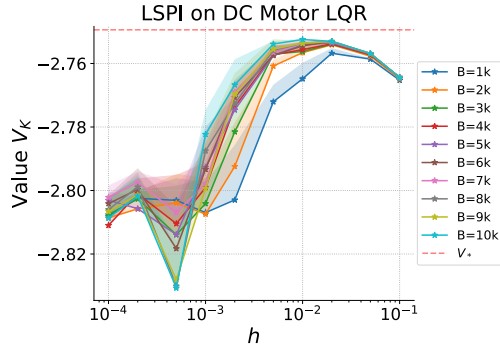

Figure 8: The trade-off in the value (higher is better) of controller $K$ learned by LSPI

## 5.2 COMPARING THE VALUE ESTIMATION ACCURACY OF TD AND MONTE-CARLO

To gain more insights into the value estimation accuracy of TD and MC, we evaluate the MSE of TD with multi-step updates, and compare it against both MC with the same $T$ and the theoretically optimal MSE* that MC could achieve, in Figure 7. The optimal MC performance is obtained by optimizing the expression of its MSE w.r.t both $T$ and $h$, which occurs at $T \approx 52$. The results show TD outperforms the optimal MC performance. This demonstrates that, when appropriately tuned, TD is a highly effective method for value estimation.

## 5.3 STOCHASTIC TD

Figure 9 illustrates that stochastic TD exhibits the same trade off with respect to the discretization step-size $h$ in both the offline and online settings. In the offline setting (a), an "epoch" refers to one complete pass over the fixed dataset. In contrast, the online setting (b) operates with streaming data, where each step corresponds to an update using a sampled transition obtained at the current step. For each $h$, we perform a grid search over constant learning rates with iterate averaging, and report the MSE of the estimate corresponding to the best hyperparameters, averaged over 50 runs.

## 5.4 STOCHASTIC CONTROL IN LINEAR QUADRATIC SYSTEMS

To test whether the trade-off occurs beyond the prediction problem, we consider the control of a continuous-time stochastic LQR, modeling a DC motor, adapted from Lewis et al. (2012) (details in Appendix A.9). The objective is to find a linear controller $K$ such that the control $\mathbf{u}(t) = K\mathbf{x}(t)$ maximizes the infinite-horizon discounted value $V_K$. Following the experimental setup of Tu & Recht (2018), we collect offline data by running a Gaussian policy $\mathbf{u} \sim \mathcal{N}(0, I_2)$ for 500 episodes of length 20 (10k samples in total, then sub-sampled to simulate smaller budgets). We then apply Least-Squares Policy Iteration (LSPI) (Lagoudakis & Parr, 2003; Tu & Recht, 2018), which alternates TD value estimation and policy improvement, yielding a controller $K$ for each step size $h$ and budget

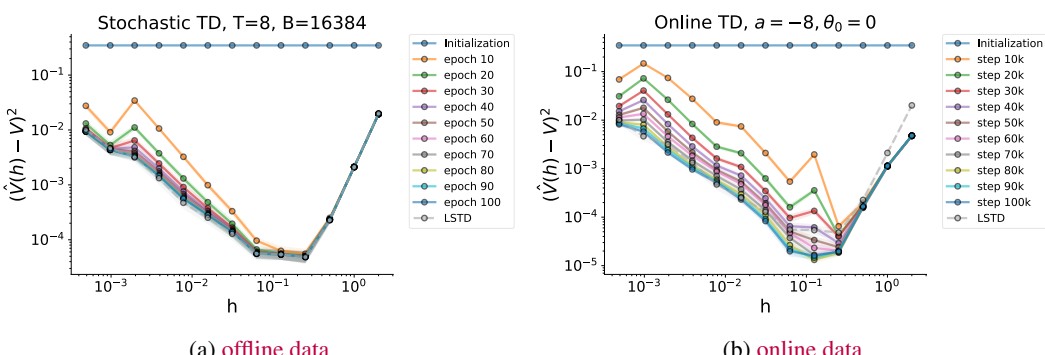

(a) offline data

(b) online data

Figure 9: MSE of Stochastic TD(0) as a function of $h$ for different numbers of updates

$B$. Each setting is repeated over 50 runs, and we report the median performance $V_K(h)$ (higher is better; optimal value $V_*$), following the evaluation protocol in Tu & Recht (2018). The shaded region covers median-60th percentile range. Figure 8 shows a clear trade-off in control performance with respect to $h$, where the optimal step size $h^*$ remains largely insensitive to the data budget $B$, mirroring our findings in the TD prediction setting.

## 6 LIMITATIONS AND FUTURE WORK

While our work provides a framework for understanding the impact of temporal resolution in TD, it has a limited scope. Our theoretical analysis of the trade-off is restricted to ~~one-dimensional~~ Langevin systems and the offline mean-path semi-gradient TD(0) algorithm. ~~As a result, the extent to which our findings generalize to more complex dynamical systems and alternative TD algorithms remains an open question.~~ Although we empirically observed a trade-off in stochastic TD in both online and offline settings, as well as in stochastic control using LSPI, it remains unclear how broadly these findings generalize to more complex dynamical systems and reward functions. Exploring how temporal resolution influences value estimation in higher-dimensional, nonlinear environments ~~and different learning paradigms~~, is an important direction for future work.

## 7 CONCLUSION

In this work, we provided a theoretical and empirical investigation into the impact of temporal resolution on offline Temporal Difference value estimation. By analyzing the Mean-Squared Error of the mean-path semi-gradient TD(0) algorithm in continuous-time stochastic linear quadratic systems, we demonstrated the existence of a non-trivial trade-off in step size $h$ where an optimal discretization improves estimation accuracy. Our analysis further revealed that unlike Monte Carlo estimation, where the optimal $h$ scales polynomially with the data budget $B$ (Zhang et al., 2023), the optimal $h$ for TD remains largely invariant to $B$. This provides practical guidance: one can select an appropriate temporal resolution under small data budgets without re-tuning for larger data.

Through extensive numerical experiments, we verified our theoretical predictions and explored the behavior of TD estimation across various parameters and algorithmic settings, including offline mean-path TD, offline stochastic TD, and online stochastic TD. Additionally, we compared TD with MC and showed that TD can outperform MC under the same data budget. Finally, we demonstrated in a stochastic control setting that the step-size trade-off persists in policy learning using LSPI.

This work establishes a framework for analyzing the role of temporal resolution in TD methods, contributing to a deeper understanding of how step size influences learning dynamics. Future directions include extending this analysis to more complex environments, higher-dimensional systems, and alternative TD formulations.

### REPRODUCIBILITY STATEMENT

The assumptions underlying our theoretical results are stated in the main text, and complete proofs are provided in the Appendix. The supplementary materials contain the Mathematica scripts and

data used for symbolic computations supporting our analysis of one-step and multi-step MSE. To illustrate the complexity of the expressions, we also provide the exact formula for the one-step MSE. In addition, we include the Python code used to conduct the offline TD numerical experiments.

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
