# OpenReview forum: "The Effect of Temporal Resolution in Offline Temporal Difference Estimation"
_ICLR.cc/2026/Conference — Submitted to ICLR 2026_

### Official Review · Reviewer_SPBP · 2025-10-27

**Soundness:** 3
**Presentation:** 2
**Contribution:** 2
**Rating:** 2
**Confidence:** 3

**Summary:**

The paper studies the impact of temporal resolution on Temporal Difference (TD) learning algorithms. The contributed theoretical analysis, focused on the continuous-time linear quadratic setting, is rigorous and provides a clear, non-trivial result: the existence of an optimal discretization step h* that minimizes the mean-squared error and its insensitivity to the data budget B, different from prior results for Monte Carlo methods. The paper is well-structured and the numerical experiments adequately support the theoretical claims.

**Strengths:**

+this paper tackles an interesting and underexplored problem-understanding the impact of temporal discretization parameter h on TD learning
+this paper derives the MSE for offline mean-path TD(0) as a function of h and characterizes the optimal h*
+comparison with the Monte Carlo results from Zhang et al. (2023) demonstrates the difference between the two estimation algorithms
+solid numerical experiments validate the theoretical findings

**Weaknesses:**

-this paper analyzes the offline mean-path semi-gradient TD(0) algorithm. This algorithm computes the exact expected update over the entire offline dataset at each step. Although this is a standard theoretical simplification (as in Bhandari et al., 2018), it sidesteps the variance introduced by stochastic sampling in practical TD(0)
-the analysis in Section 5 numerically shows that the algorithm converges to the LSTD solution (Eq. 10). However, the paper would be strengthened by a formal discussion of the assumptions required for this.
-the suggestion in Section 4.2 for a burn-in phase to find h* is viable. However, it relies on having a Monte Carlo estimate of the true value V to compute the empirical MSE. In most real-world problems, V is unknown, which is exactly why we use TD algorithms in the first place. Put differently, if one already has a good enough estimate of V to perform this tuning, why need TD?
-the constants invovled in the optimal h* are complicated functions of the underlying unknown system parameters, and it is inviable to get h* without knowing the dynamics.
-The analysis relies on a specific, well-chosen feature \phi(x), and it is not clear how the findings about the temporal resolution and the optimal h* depend on the given perfect representation and especially in the presence of learning approximation error?

**Questions:**

See above

---

> ### Author Response · Authors · 2025-12-04
>
> Thank you for your feedback. We address your questions and concerns below:
>
> > this paper analyzes the offline mean-path semi-gradient TD(0) algorithm. This algorithm computes the exact expected update over the entire offline dataset at each step. Although this is a standard theoretical simplification (as in Bhandari et al., 2018), it sidesteps the variance introduced by stochastic sampling in practical TD(0)
>
> We added justifications for mean-path and offline setting in Section 3.2 and 3.3. And we have also added experimental results (Fig. 9) showing that the trade-off persists in stochastic TD. In addition, we have added finite time bounds for stochastic TD under the continuous and unbounded state space of LQ systems in Section 4.4 and Appendix A.7.
>
> > the analysis in Section 5 numerically shows that the algorithm converges to the LSTD solution (Eq. 10). However, the paper would be strengthened by a formal discussion of the assumptions required for this.
>
> We added the analysis on the sufficient conditions for convergence to Section 4.4, Appendix A.4.1, and A.5.
>
> > it is not clear how the findings about the temporal resolution and the optimal h* depend on the given perfect representation and especially in the presence of learning approximation error?
>
> We added analysis showing that MSE scaling retains the order w.r.t h under function approximation errors, in Section 4.3.

---

### Official Review · Reviewer_1VwY · 2025-10-30

**Soundness:** 3
**Presentation:** 3
**Contribution:** 2
**Rating:** 2
**Confidence:** 4

**Summary:**

The paper analyzes how temporal discretization affects TD(0) value estimation for continuous-time linear quadratic systems. Main claim is there's an optimal discretization step h*. Specific special cases are analyzed.

**Strengths:**

The question is interesting - temporal resolution is overlooked in RL so studying it systematically makes sense.

The empirical results clearly show the U-shaped curve for optimal h, which is the main point.

The comparison showing TD has constant h* while MC has h* ~ B^1/2 is a neat observation.

**Weaknesses:**

The scope is way too narrow. This is literally just:

1D Langevin dynamics
One specific algorithm (mean-path TD, not even standard TD)
Quadratic features with zero approximation error
Offline only

How is anyone supposed to know if this generalizes? Linear quadratic systems are the simplest possible case. The authors admit this in Section 6 but it's a dealbreaker. Why not test on at least a 2D system or some standard RL benchmark before submitting?
Mean-path TD is non-standard. It updates using the mean gradient over the whole dataset (eq 4), not stochastic updates like regular TD. This is basically a batch method. So you're analyzing convergence to LSTD, not how good LSTD is, and this isn't how people actually use TD in practice.

The practical advice doesn't work. The optimal h* in Corollary 4.2 depends on constants C11, C12, etc that require knowing system parameters. But in real RL you don't know these! They suggest doing grid search over h with a small budget, but that means collecting data at multiple frequencies which is wasteful. And if you don't have a model, how do you know your estimated h* is any good?

The MC comparison is weird. Figure 6 claims TD "outperforms" MC but they use different horizons (T=8 for TD vs T≈52 for MC). That's not a fair comparison - you're giving the methods different amounts of information. Also MC is known to have high variance for infinite horizon so this isn't surprising.

Experiments only test the same 1D Langevin system with different parameters. No other dynamical systems, no higher dimensions, no standard benchmarks. The shaded regions in Figures 4-6 show pretty large variance too.

**Questions:**

For standard stochastic TD(0) (not mean-path), would you see the same behavior?

How sensitive is this to the feature choice? What if φ(x) was different?

Can you quantify *when* the B terms in eq 8 become negligible?

Why not test even just 2D Langevin to show some generalization?

This reads more like preliminary work that needs extension before publication. Can you could show similar tradeoffs exist in more realistic settings, or give better practical guidance?

---

> ### Author Response · Authors · 2025-12-04
>
> Thank you for your feedback. We address your questions and concerns below:
>
> > For standard stochastic TD(0) (not mean-path), would you see the same behavior?
> > Can you could show similar tradeoffs exist in more realistic settings
>
> Yes. We added experimental results (Fig. 9) showing that the trade-off persists in stochastic TD.
>
> > How sensitive is this to the feature choice? What if φ(x) was different?
>
> We added analysis showing that MSE scaling retains the order w.r.t h under function approximation errors, in Section 4.3;
>
> > Why not test even just 2D Langevin to show some generalization?
>
> We added experimental results (Fig. 4) showing that the trade-off persists in three-dimensional systems.
>
> >  One specific algorithm (mean-path TD, not even standard TD) Quadratic features with zero approximation error Offline only
>
> We added justifications for mean-path and offline setting in Section 3.2 and 3.3.
>
> > The MC comparison is weird. Figure 6 claims TD "outperforms" MC but they use different horizons (T=8 for TD vs T≈52 for MC). That's not a fair comparison - you're giving the methods different amounts of information. Also MC is known to have high variance for infinite horizon so this isn't surprising.
>
> T=52 is the optimal horizon that minimizes the closed-form MSE for MC in this instance. We are giving extra information to MC here, thus biasing the comparison in favor of MC. The fact that TD still outperforms the optimal MC highlights the efficiency of TD  in this instance.
> We respectfully note that a higher variance does not necessarily imply a higher MSE since MSE is determined by both bias and variance. It is therefore not a priori obvious that TD will outperform MC.

---

### Official Review · Reviewer_j8yX · 2025-10-31

**Soundness:** 3
**Presentation:** 4
**Contribution:** 3
**Rating:** 8
**Confidence:** 3

**Summary:**

This paper focuses on continuous time decision making problems and tries to find the optimal temporal resolution to sample such a system to minimize the mean squared error (MSE) of a temporal difference (TD) based value estimator in the offline setting (where the data has been sampled upfront before value estimation begins). To do so it focuses on a class of problems that can be analyzed theoretically, namely continuous-time stochastic linear quadratic systems. The paper analyzes how the MSE of this offline semi-gradient TD algorithm with respect to the true value function changes as a factor of all the different variables in this system, and then characterizes the optimal temporal resolution for a given system. While the optimal temporal resolution would require knowing system dynamics upfront or evaluating multiple different resolutions, the paper proposes a practical approach to find the right temporal resolution by using a smaller batch based on the theoretical finding that the MSE is independent of batch size if it is large enough.

Finally, the paper validates its results and additional questions that the analysis brings up by evaluating a particular linear quadratic system with different parameters.

**Strengths:**

* The paper sets up a clear problem statement and a valid simplification that can be analyzed
* The theorems appear to be correct, although this reviewer mostly skimmed the proofs
* The paper presents the results of the analysis in a manner that is mostly easy to parse, specifically equation 7 and the subsequent explanation. These efforts will make it easy for an intelligent reader who is not immersed in the literature this paper builds upon to follow the paper and understand it.
* Subsection "How to choose temporal resolution for TD" is a good example of practical takeaways that this paper seeks to communicate.
* Deeper comparison to closest comparable work that this paper seems to build upon in Section 4.3 is also appreciated.
* The empirical evaluation seems to bear out the theoretical analysis presented in the paper.
* The presentation of the empirical evaluation is also very clear and well communicated.

**Weaknesses:**

* While it is understandable that the theoretical analysis requires a system like a linear quadratic one, it would be good to evaluate empirically if the findings hold for others.
* A practical example of whether the strategy suggested at the end of section 4.2, perhaps paired with an unknown dynamical system as mentioned in the previous point, would be useful.
* A little more detail in the captions for the Figures 2-6 would be helpful for a reader.
* Minor: figure 3 is placed after figures 4 and 5 in the paper

**Questions:**

For the result in Figure 6, since MC MSE is dependent on batch size B, if we increased B would we see it get closer to TD? Or would the improved estimation of TD outpace improvements to MC?

---

> ### Author Response · Authors · 2025-12-04
>
> Thank you for the feedback.
> In the revision, we have made several additions to improve the technical depth of the work, as summarized in our general response to all.
>
> To answer your question, increasing B reduces the MSE for both MC and TD. MSE for both algorithms involves terms that scale with 1/B. In the limit, i.e. when B is large, these terms vanish and the performance comparison depends on the parameters. Consequently, it is difficult to provide a universal answer regarding whether MC 'gets closer' to TD in all settings.
> However, comparing the relative performance at a specific B is not the primary objective of this paper. Rather, our main focus is to characterize the trade-off regarding the temporal resolution in TD learning.

---

### Official Review · Reviewer_XWV6 · 2025-11-01

**Soundness:** 3
**Presentation:** 3
**Contribution:** 2
**Rating:** 4
**Confidence:** 3

**Summary:**

The paper analyzes how temporal resolution (step size $h$) impacts offline value estimation with mean-path semi-gradient TD(0) on continuous-time linear systems with quadratic rewards. For fixed data budget $B$, they derive a small-$h$ MSE expansion, $\mathrm{MSE}_t \approx C_0 + C_1 h + C_2 h^2$, implying an interior optimum $h^{\star}>0$; unlike Monte Carlo, where $h^{\star}$ scales with $B$. Experiments show TD iterates reach the LSTD fixed point, MSE is minimized at intermediate $h$, and the empirical $h^{\star}$ is largely insensitive to $B$.

**Strengths:**

- Focus on a concrete and important knob—temporal resolution—for offline TD value estimation.
- Tractable MSE expansion in $h$ with identifiable coefficients, yielding a closed-form $h^{\star}$.

**Weaknesses:**

- Results are for 1D Langevin/LQ with linear features that exactly span the value; unclear whether the same $h^{\star}$ behavior holds with function-approximation error, higher-dimensional or nonlinear dynamics, or non-quadratic rewards.
- Mean-path semi-gradient TD(0) is analytically friendly but less standard than stochastic TD, TD($\lambda$), or GTD

**Questions:**

- Do you expect similar $h^{\star}$–vs–$B$ insensitivity for stochastic TD with constant stepsizes and iterate averaging, or for TD$(\lambda)$?

---

> ### Author Response · Authors · 2025-12-04
>
> Thank you for the feedback. In the revision, we have made the following additions to address your concern:
> 1. added analysis showing that MSE scaling retains the order w.r.t h under function approximation errors;
> 2. added experimental results showing that the trade-off persists in higher-dimensional systems;
> 3. added justifications for mean-path setting;
> 4. added finite time bounds for stochastic TD under the continuous and unbounded state space of LQ systems;
> 5. added experimental results showing that the trade-off persists in stochastic TD;
>
> Regarding your question on whether the h*-vs-B relationship holds for stochastic TD with constant learning rate and iterate averaging, the answer is yes, as demonstrated in the added Fig. 9.

---

### Author Response · Authors · 2025-12-03
**Summary of Revisions**

We would like to thank all reviewers for their time and effort in reviewing our paper.

We have uploaded a revision of the paper that incorporates significant additions to both the theoretical analysis and empirical results, addressing the feedback. All changes are marked in **purple**. We would like to summarize the key additions below:

**Theoretical Analysis:**

1. Added discussions on the offline setting to Section 3.2 and the mean-path setting to Section 3.3 (Reviewer XWV6, 1VwY)
2. Added analysis about offline mean-path TD under function approximation error in Section 4.3 (Reviewer XWV6, 1VwY, SPBP)
3. Added convergence analysis in Section 4.4 covering three regimes:
    - convergence of offline mean-path (Reviewer SPBP)
    - finite sample analysis of online mean-path
    - finite sample analysis of stochastic TD in the online setting

**Empirical Results:**

1. Added experiments on 3-dimensional system (Section 5.1, Fig. 4), confirming that the trade-off and dependence of h* on B generalizes to the vector case (Reviewer XWV6, 1VwY)
2. Added experiments on stochastic TD (Section 5.3, Fig. 9), demonstrating that the trade-off persists in the practical setting (Reviewer XWV6, SPBP)
3. Added experiments on stochastic control using Least Squares Policy Iteration (Section 5.4, Fig. 9), showing that the trade-off persists in the control setting.

**Sections 1, 6, and 7** have been updated to reflect these additions.

We believe that these updates in the revision have substantially strengthened the paper.

---

> ### Author Response · Authors · 2025-12-04
>
> Note: we recognize that the window for discussion has effectively closed. However, we feel it is important to submit our detailed responses to honor the time and effort the reviewers and ACs have invested in evaluating our work. We have dedicated significant effort during the rebuttal phase to incorporate your feedback into the revisions. We hope that the improved quality of the paper speaks for itself, even in the absence of back-and-forth.

---

### Meta-Review · Area_Chair_cWpf · 2025-12-26

**Summary:**

The decision is driven by persistent concerns regarding scope, generality, and practical relevance, which remain despite the authors’ substantial revision effort. Multiple reviewers emphasized that the paper’s theoretical and empirical results are confined to a highly restricted setting: continuous-time linear quadratic systems, largely perfect feature representations, and a specific algorithmic variant (offline mean-path semi-gradient TD). While analytically tractable, this setting is viewed as too specialized to support broad conclusions about temporal resolution in TD learning more generally. A recurring concern is that the analysis focuses on mean-path (batch) TD and convergence to LSTD, rather than stochastic TD as commonly used in practice. Although the authors added stochastic TD experiments and bounds, reviewers remained unconvinced that these additions sufficiently bridge the gap between the theoretical model and realistic RL workflows, where sampling noise, partial observability, and non-linear function approximation dominate. Overall, while the reviewers generally agreed that the technical analysis is careful and correct within its assumptions, there was broad skepticism that the contribution rises to the level required for acceptance.

**Reviewer Concerns:**

The authors made a strong effort to address reviewer feedback by adding analyses for function approximation error, stochastic TD, and higher-dimensional systems, as well as finite-sample convergence results. These additions improved the completeness and rigor of the manuscript. The revised manuscript provides clearer justification for the use of mean-path TD and the offline setting, and explicitly situates these choices within the theoretical literature. Added experiments demonstrate that the key U-shaped trade-off persists across several controlled variations, strengthening internal consistency.
Despite added experiments, the work remains confined to linear quadratic systems and carefully structured settings. Reviewers were not persuaded that the findings generalize to the broader class of problems where TD learning is typically applied. The method for identifying h* continues to rely on assumptions that are unrealistic in many RL applications, such as access to accurate value estimates or repeated data collection at multiple temporal resolutions. Several reviewers viewed the paper as a technically solid but incremental extension of existing analyses, with limited conceptual or algorithmic innovation beyond a narrow setting.

**Reviewer Scores:**

Reviewer 1VwY (Initial: 2 – Reject)
Despite the added experiments and analysis, this reviewer would likely remain at reject, as the fundamental concerns about scope, realism, and generalization persist.

Reviewer SPBP (Initial: 2 – Reject)
While acknowledging improved rigor, this reviewer would likely maintain a reject recommendation due to unresolved concerns about practicality and dependence on strong assumptions.

Reviewer XWV6 (Initial: 4 – Marginal Reject)
Might modestly soften their stance in light of the added stochastic TD analysis, but would likely remain below the acceptance threshold.

Reviewer j8yX (Initial: 8 – Accept)
This reviewer was positive and would likely remain supportive; however, their view appears to be an outlier relative to the broader set of reviews.

---

### Decision · Program_Chairs · 2026-01-26

Reject